# Mitigation of Aerosol and Microbial Concentration in a Weaning Piggery by Spraying Nanobubble Ozone Water with an Ultrasonic Sprayer

**DOI:** 10.3390/ani14050657

**Published:** 2024-02-20

**Authors:** Takumi Yoshino, Atuso Ikeguchi

**Affiliations:** 1Graduate Schools, Utsunomiya University; 350 Minemachi, Utsunomiya 321-8505, Japan; 2Department of Environmental Engineering, Faculty of Agriculture, Utsunomiya University, 350 Minemachi, Utsunomiya 321-8505, Japan; ike14000@gmail.com

**Keywords:** aerosol, airborne microbial, ozone solution, ultrasonic sprayer, weanling piggery

## Abstract

**Simple Summary:**

Elevating biosecurity in livestock housing is vital in ensuring the safety of animal products. In Japan, approximately 70% of post-weaning mortality is attributable to respiratory pathogens. Microorganisms, including viruses and bacteria, do not float independently but depend on aerosols for transmission. Hence, controlling aerosols is crucial in reducing pathogens and enhancing livestock production. This study aimed to decrease aerosol and microbial concentrations, along with airborne viruses, by employing ozone solution spraying. Experiments were conducted in an integrated management farm in Fukushima Prefecture, involving nanobubble ozone water sprayed with an ultrasonic sprayer. Results showed a substantial reduction in microbial concentrations and aerosol mass, indicating its effectiveness.

**Abstract:**

Enhancing biosecurity measures in livestock is an essential prerequisite for producing animal products with the highest levels of safety and quality. In Japan, 70% of the mortalities post-weaning are attributed to respiratory pathogens. The research has shown that microorganisms, including both viruses and bacteria, do not merely float in the air independently. Instead, they spread by adhering to aerosols. Therefore, improving the control of aerosol dissemination becomes a critical strategy for reducing pathogenic loads and boosting the overall efficiency of livestock production. This study focused on reducing concentrations of aerosol particles, airborne microbial concentrations, and airborne mass concentrations by spraying ozone solution with an ultrasonic sprayer. The experiments were conducted at a farm in Fukushima Prefecture, Japan, known for its integrated management system, overseeing a herd of 200 sows. Nanobubble ozone water particles were dispersed using an ultrasonic sprayer, which allowed the particles to remain airborne significantly longer than those dispersed using a standard nozzle, at a rate of 30 mL per weaning pig 49 days old, for a 10 min period. This procedure was followed by a 10 min pause, and the cycle was repeated for 17 days. Measurements included concentrations of airborne bacteria, aerosol mass, and aerosol particles. The findings demonstrated a substantial reduction in airborne microbial concentrations of *Escherichia coli* and *Staphylococcus aureus* in the treated area compared to the control, with reductions reaching a peak of 85.7% for *E. coli* and 69.5% for *S. aureus*. Aerosol particle sizes ranging from 0.3–0.5 µm, 0.5–1.0 µm, 1.0–2.0 µm, 2.0–5.0 µm, to 5.0–10.0 µm were monitored, with a notable decrease in concentrations among larger particles. The average aerosol mass concentration in the test area was over 50% lower than in the control area.

## 1. Introduction

A hygienic environment must be maintained in livestock housing for livestock productivity. In Japan, 70% of the mortalities post-weaning are attributed to respiratory pathogens [1]. The two main types of environmental elements in barns are the thermal environment and the air hygiene environment. Environmental factors in the thermal environment include temperature, humidity, and radiation, while the air quality environment includes various gases, aerosols, and microbes. In the air hygiene environment, odorous gases and airborne microorganisms are factors in odor problems and livestock infectious disease issues. Odor problems account for the largest proportion of complaints about livestock operations [2] and have become more apparent in recent years as livestock operations have become larger and more integrated with residential areas.

Aerosols are defined as tiny liquids or individual particles that float in the air [3,4]. A human produces 1 million aerosols/fraction in each cough or sneeze. In the context of aerosol science and infectious disease transmission, bioaerosols comprising droplets expelled by human respiratory activities, including exhalation, sneezing, and coughing, can be vectors for airborne pathogens. When inhaled by susceptible individuals, these particulate matter carriers facilitate the propagation of a spectrum of infectious diseases, notably influenza, measles, swine flu, chickenpox, smallpox, and severe acute respiratory syndrome (SARS), among others [5].

Aerosols in livestock housing are caused by livestock feces, body hair, droplets, and feeds. In particular, in the piggery house, a previous study has revealed that the aerosol concentration is between 1.3 and 23.5 mg/m^3^, and 80–90% of the aerosols in piggeries originate from the feed, 2–8% from manure, and 2–12% from the pigs’ bodies. Because the temperature, humidity, and activity of livestock originate aerosols, the concentration of aerosols is linked to animals’ actions. About 87% of the composition of aerosols in the piggery is dried goods, of which about 24% is protein, 4% is fat, and 15% is mineral content [6]. Ikeguchi (2001) showed that the peak of the aerosol concentration is closely related to the operation inside the livestock housing, such as feeding and removing waste [7]. The aerosol concentration becomes high just after the operation comes into practice.

The microbes in the air does not float on their own but attach to aerosols and spread inside livestock housing [8,9]. As microbe and aerosol particles are strongly correlated, the effective mitigation of aerosol particles is anticipated to be directly associated with a consequent reduction in microbial populations. Elevated concentrations of fine aerosol particles can permeate deeply into the bodily tissues of both livestock and farm personnel via respiration. Therefore, these aerosols harbor the potential to induce respiratory ailments, cardiovascular disorders, type two diabetes, and, in severe instances, malignancies. Consequently, the mitigation of such aerosol particles is very important in efforts aimed at enhancing indoor air quality [10,11,12,13,14,15].

Technologies to reduce the aerosols and microbes inside livestock housing include electrostatic precipitators, spraying with weak acidic water, and photocatalytic ventilation systems. An electrostatic precipitator is a dust collection method that uses electrostatic force, in which a high DC voltage generates a corona discharge to electrically charged particles, such as aerosols in the air, and collects these charged particles with electrostatic force. This technology is also applied in home air purifiers. When operated in a container-type weaning piggery (12 × 2.2 × 2.2 m), the aerosol particle concentrations decreased by 70–80% in every diameter range, and airborne microbial concentrations also declined by around 80% [16].

The main ingredient in the spray of weak acidic water is a solution containing hypochlorous acid, which has sterilizing effects in the range of pH 5.0–6.0. Since it does not contain any substance that is harmful to mammals, the solution is used at food factories, hospitals, and nursing homes. A spraying experiment in a container-type weaning piggery (11.9 × 2.2 × 2.15 m) achieved a decrease of approximately 50% of aerosol mass concentration after spraying the solution at 9.69 g/m^3^∙min for 21 min [16].

Photocatalyst is the generic term for substances that exhibit catalytic activity when irradiated with light and can inactivate microbes, including *Escherichia Coli* and *Staphylococcus aureus*. This technology decreased the microbe when installed in the livestock house on an experimental basis [16].

Another way to decrease aerosol particle concentration is feeding liquid feed, which is either fully liquid or has a high moisture content [17].

In the previous study by our research group, experiments on decreasing aerosols were conducted [18]. In an animal experimental facility infected with porcine saperovirus (PSV), porcine epidemic diarrhea virus (PEDV), and coliforms, an acidic water was sprayed weakly at 100 mL/min for 10 min, which was sprayed with 1000 ml of sterile phosphate saline solution. This reduced the rate of aerosol concentration compared to the no-spray condition.

Spraying experiments were also carried out on livestock sites. The weak acidic water, with a diameter of 200 µm at 1000 mL, was sprayed for 15 min for 2 days in an integrated operation house located in Tenei-mura, Fukushima Prefecture, during the summer, autumn, and winter seasons. Spraying the solution inside the livestock house decreased the aerosol mass concentration on average by 17% in summer, 22.3% in autumn, and 10.7% in winter.

In this experiment, nanobubble ozone water was chosen as the disinfectant solution. This innovative approach sets it apart from traditional ozonated water, as it features ozone dissolved as nanoscale particles within the solution, tailored specifically for agricultural applications. Furthermore, while maintaining effective bactericidal properties, this method presents a significant cost benefit by enabling on-site production at the farm, distinguishing it from conventional disinfectants.

Hence, the objective was to spray a solution to mitigate aerosols, thereby enhancing biosecurity measures in livestock, which is crucial in ensuring the production of animal products that meet the highest safety and quality standards.

## 2. Materials and Methods

### 2.1. Weaning Piggery

The research was conducted in an integrated operation with 200 sows in Fukushima Prefecture, Japan. The weaning piggery has a ventilation system, with a total of four rooms and four pens per room. The volume of each room was 117.6 m^3^, and the area of the pen was 9.87 m^2^. The floor of the stalls was made of concrete and slatted flooring, and each room was equipped with two ventilation fans and an inverter controlled by temperature. When the research began, the experimental group comprised 80 pigs, and the control group consisted of 88 pigs, with both groups being 49 days old. Figure 1 and Figure 2 show the plan view of the inside of the experimental farm and the measurement and spray points.

### 2.2. System for Spraying

Continuous spraying was conducted as the spraying and interval times were implemented, with intervals meticulously set at 10 min increments. Whenever the relative humidity within the facility exceeded 80%, the spray was stopped to avert potential adverse repercussions on the weaning pigs. Furthermore, a comparative analysis was undertaken, setting specific areas as the test and control regions, encompassing four pig stalls within a single enclosure. The whole system is illustrated in Figure 3, and a photograph of the equipment is in Figure 4. Nanobubble ozone water, demonstrated to be non-harmful to mammals in experimental studies and exhibiting a high bactericidal effect, was chosen as the spray solution. This solution was produced in the spraying system. The solution was spread from a height of 2.6 m below the ceiling using an ultrasonic sprayer, which produces particles with diameters ranging from 5 to 10 μm. This method was selected to ensure that the particles remain suspended in the air for an extended period. Additionally, this approach is more cost-effective compared to conventional nozzles. Precise control over the spray regimen was maintained by factoring in time, humidity levels, and aerosol concentrations. Figure 5 shows an image during spraying.

### 2.3. Measurement Items

Between 19 May and 6 June 2023, a series of measurements were conducted to assess airborne microbial concentrations, aerosol particle concentrations, and aerosol mass concentrations. Each set of measurements was replicated three times to ensure accuracy and reliability.

#### 2.3.1. Measurement of the Aerosol Particle Numbers

The aerosol particle numbers were measured by the Optical Particle Sizer (OPS) Model 3330 (TSI Inc., Shoreview, MN, USA) [19,20,21,22]. The OPS counted the aerosol particle numbers in up to 16 separate channels with a 1 s time resolution and <3000 aerosol particle number concentrations/cm^3^. The aerosol particle numbers were divided into five size fractions: 0.3–0.5 μm, 0.5–1.0 μm, 1.0–2.0 μm, 2.0–5.0 μm, and 5.0–10.0 μm (recommended by the International Organization for Standardization) [23]. During the experiment, the OPS sampled air at a rate of 1.0 L/min with a ±5% variance. The OPS was programmed to operate in repeat intervals of 5 min, in which aerosol particle measurements were conducted for 1 min, followed by a suspension period of 4 min. The total number of aerosol particles was expressed in particles/m^3^. The particle concentration was measured to determine each range of aerosol diameter during the experiment.

#### 2.3.2. Calculation of Aerosol Mass Concentration

The aerosol mass concentration was measured by the high-volume sampler (HV500; SIBATA, Japan). The air was sampled by PM.10, which aspirated the particles floating in the air smaller than 10.0 μm diameter. The glass fiber filter was attached to a high-volume sampler and aspirated the air at a flow rate of 500 L/min for 10 min.

The filter was subjected to a drying process at 100 °C for one hour in a drying oven, both pre- and post-experimentation. The aerosol mass concentration was subsequently calculated based on the weight differential of the filter. The mass concentration was measured to determine the total amount of aerosols during the experiment.

#### 2.3.3. Collection, Culturing, and Counting of Different Types of Airborne Bacteria

In this study, different types of airborne bacteria were collected using a liquid cyclone air sampler (Coriolis μ; Bertin Inc, Montigny-le-Bretonneux, France) [19,20,22,24,25]. Air samples were collected over a 20 min period, each in a 10 mL phosphate-buffered saline solution at a flow rate of 300 L per minute, and promptly transported to the laboratory in a temperature-controlled container for immediate processing. Employing the 10-fold serial dilution method, a key approach in microbiological analysis, 1 mL aliquots from these samples were diluted with 9 mL of sterile diluent, resulting in sequential dilutions (0.1 M, 0.01 M, and 0.001 M). These diluted samples were then inoculated onto cultural medium sheets (Sanita-kun; JNC Corporation, Tokyo, Japan) in triplicate for each dilution, totaling nine sheets per bacterial type. Following inoculation, the sheets were incubated at 35 °C for 48 h for Aerobic microorganisms and 24 h for *E. coli* and *S. aureus*. Airborne bacterial concentrations were ascertained through colony enumeration post-incubation, employing the standard plate count method in accordance with JNC Corporation guidelines [19,20,22]. Among the dilutions, culture medium sheets showing 30–300 colonies were used to calculate the concentrations of various types of airborne bacteria using Equation (1) [22,26]. The principal detection limit of the culture medium sheet method is one colony on each of two culture medium sheets among three culture medium sheets with 0.1 M diluted sample after incubation. The total number of colonies was expressed as CFU/m^3^.
(1)$$\begin{array}{c} C=\log_{10}\left(\frac{N\times 10^{n}}{V_{p}}\times V_{S}\times \frac{1}{V_{a}}\right) \end{array}$$
where C is the airborne bacteria concentration (log_10_ CFU/m^3^); *N* is the number of colonies on a culture medium sheet (30–300 colonies); *n* is the serial dilution factor (*n* = 1 for 0.1 M dilution, etc.); $V_{p}$ is the sample volume cultured (1 mL in this study); $V_{s}$ is the total volume of stock sample used for culture (1 mL in this study); and $V_{a}$ is the total volume of air sampled using Coriolis μ (3 m^3^ in this study).

### 2.4. Statistical Analysis

In this study, the statistical analysis was conducted using RStudio, a comprehensive statistical computing and graphics environment. To compare the means of two independent groups, a two-sample *t*-test was employed for all of the results.

## 3. Results and Discussion

### 3.1. Results concerning the Experimental Environment

The results found for the piggery environment are provided in Figure 6. The average temperature during the experiment was 20.4 °C in the test area and 22.6 °C in the control area. Significant differences observed in temperature indicated that the test area was slightly cooler than the control area. The temperature was likely influenced by the evaporation of the liquid sprayed in the test area, which caused the temperature to decrease.

Figure 7 shows the results of relative humidity in the test and control areas. The average humidity was 71.0% in the test area and 63.5% in the control area. The test area was wetter than the control area because of the spraying solution in the test area. Significant differences were observed in comparing both areas.

The temperature humidity index (THI), which is a measure used to assess the combined effects of temperature and humidity on animals, particularly livestock, is in Figure 8. The THI is crucial for evaluating environmental stress, such as heat stress, which can significantly impact animal welfare and productivity. The average THI during the experiment was 66.9 in the test area and 69.7 in the control area.

### 3.2. Airborne Microbial Concentrations

Before spraying, airborne microbial concentrations of Aerobic microorganisms, *E. coli,* and *S. aureus* were 3.33 log_10_ CFU/m^3^, 2.88 log_10_ CFU/m^3^, and 3.05 log_10_ CFU /m^3^, respectively. On day four, 22 May (52 days old), the concentrations of Aerobic microorganisms, *E. coli*, and *S. aureus,* were 3.54 log_10_ CFU m^3^, 2.82 log_10_ CFU /m^3^, and 2.96 log_10_ CFU/m^3^, respectively, in the control area, and 2.95 log_10_ CFU/m^3^, 1.96 log_10_ CFU/m^3^, and 2.40 log_10_ CFU/m^3^ in the test area. Significant reductions of 85.7% for *E. coli,* 75.4% for Aerobic microorganisms, and 71.3% for *S. aureus* were observed in the test area compared to the control area (Figure 9).

On day eleven, 29 May (59 days old), the concentrations of the Aerobic microorganisms, *E. coli,* and *S. aureus* were 3.24 log_10_ CFU/m^3^, 3.09 log_10_ CFU/m^3^, and 3.22 log_10_ CFU /m^3^ in the control area, and 3.03 log_10_ CFU/m^3^, 2.68 log_10_ CFU/m^3^, and 2.68 log_10_ CFU/m^3^ in the test area, respectively. Compared to the control area, a 34.3% reduction in Aerobic microorganisms was observed, as well as a significant reduction in *E. coli* and *S. aureus* of 60.2% and 69.5%, respectively (Figure 10).

On day seventeen, 6 June (65 days old), the concentrations of Aerobic microorganisms, *E. coli,* and *S. aureus* were 3.66 log_10_ CFU/m^3^, 3.18 log_10_ CFU/m^3^, and 3.00 log_10_ CFU/m^3^, respectively, in the control area, and 2.79 log_10_ CFU/m^3^, 2.85 log_10_ CFU/m^3^, and 3.07 log_10_ CFU/m^3^ in the test area. Compared to the control area, reductions of 89.1%, 67.1%, and 62.8% were observed for Aerobic microorganisms, *E. coli*, and *S. aureus* (Figure 11).

Therefore, the test area exhibited substantial mitigation of airborne microbial concentrations, especially of *E. Coli* and *S. aureus*, with reductions reaching as high as 85.7% and 69.5%, respectively, compared to the control area.

Spraying the ozone solution at a 5–10 mm diameter using an ultrasonic sprayer reduced airborne microbial concentration, aerosol particle concentration, and aerosol mass concentration. The airborne microbial concentration in the test area was maintained lower than in the control area for 17 days. A previous study focused on the effect of different particle sizes on the number of airborne bacteria and aerosol and reported that a particle size of 100 to 200 μm was desirable [27]. The authors also found that the 200 μm sprayed particle size lasted longer than the 100 μm sprayed particle size in reducing the airborne microbial concentration [27]. In that experiment, the 200 μm particle size was more effective than the 100 μm particle size.

The larger the particle size, the larger the area of adhesion; thus, it is presumed that the sprayed particles better adhered to the airborne microorganisms and disinfected as they fell.

However, we sprayed the solution in a smaller size than the previous study, aiming to float the solution in the air for a longer time to increase the time that the aerosol attached to microbes. The results showed that a tiny particle solution can also reduce the airborne microbial concentration. The solution sprayed with an ultrasonic sprayer flows in the air for a longer time compared to the one with the normal nozzle, allowing the solution to spread throughout the livestock house. Thus, the results obtained by this research could be more effective in decreasing the aerosols and bacteria.

The ozone water decreases the airborne microbial concentration due to the sterilizing properties of ozone. Ozone functions as an efficacious oxidizing agent, capable of inducing oxidative harm to the cellular constituents of bacteria. This oxidative impairment predominantly impacts the cell membrane and essential cellular configurations, culminating in cellular death. The capacity of ozone to effectively target a wide array of microorganisms, such as *E. coli* and *S. aureus*, stems from its potential to compromise cellular integrity [28,29]. Ozone can oxidize various components of the cell envelope, including polyunsaturated fatty acids, membrane-bound enzymes, glycoproteins, and glycolipids, leading to leakage of cell contents and eventually causing lysis [30].

*E. coli*, a Gram-negative bacterium, has a thin peptidoglycan layer and an outer membrane, which includes lipopolysaccharides. *S. aureus*, a Gram-positive bacterium, has a thicker peptidoglycan layer but lacks an outer membrane. The variations in cell wall and membrane structures observed between Gram-positive and Gram-negative bacteria did not appear to be a determining factor influencing survival in the presence of ozone [31]. The ozone can penetrate these cell wall structures, causing significant damage; however, the effectiveness of ozone is influenced by these structural differences, as it can disrupt cell wall and membrane integrity.

### 3.3. Aerosol Particle Concentration

Figure 12 illustrates the diminution rates in aerosol particle concentration in both control and test areas, as observed on days 4, 11, and 17. The larger aerosol particles exhibited a more pronounced reduction than their smaller counterparts. Notably, on day 11, an increase in all particle sizes was observed in the test area, attributable to a 20% elevation in relative humidity compared to days 4 and 17, which hindered the evaporation of the solution dispersed by the ultrasonic sprayer. This phenomenon can be primarily ascribed to the higher gravitational settling rate of larger particles than smaller particles, leading to a more substantial deposition of larger-sized particles.

On day 4, May 22 (52 days old), the aerosol particle concentrations in the control area were 5.64 × 10^8^ particles/m^3^ for particles 0.3–0.5 μm in diameter, 3.91 × 107 particles/m^3^ for particles 0.5–1.0 μm, 1.81 × 10^7^ particles/m^3^ for particles 1.0–2.0, 1.77 × 10^7^ particles/m^3^ for particles 2.0–5.0 μm, and 1.02 × 10^7^ particles/m^3^ for particles 5.0–10.0 μm. However, in the test area, the concentration was 1.02 × 10^9^ particles/m^3^ for particles 0.3–0.5 μm in diameter, 7.29 × 10^7^ particles/m^3^ for particles 0.5–1.0 μm, 1.67 × 10^7^ particles/m^3^ for particles 1.0–2.0 μm, 1.27 × 10^7^ particles/m^3^ for particles 2.0–5.0 μm, and 6.64 × 10^6^ particles/m^3^ for particles 5.0–10.0 μm. There was an 81.8% significant increase, 86.6% significant increase, 7.5% decrease, 28.5% significant decrease, and 34.9% decrease in the test area compared to the control area for the particle ranges, respectively (Figure 13).

On day 11, May 29 (59 days old), the aerosol particle concentrations were 5.90 × 10^8^ particles/m^3^ for particles 0.3–0.5 μm in diameter, 3.18 × 10^7^ particles/m^3^ for particles 0.5–1.0 μm, 9.07 × 10^6^ particles/m^3^ for particles 1.0–2.0 μm, 1.04 × 10^7^ particles/m^3^ for particles 2.0– 5.0 μm, and 8.06 × 10^6^ particles/m^3^ for particle 5.0–10.0 μm in the control area. Meanwhile, in the test area, the concentration was 1.08 × 10^9^ particles/m^3^ for particles 0.3–0.5 μm in diameter, 1.95 × 10^8^ particles/m^3^ for particles 0.5–1.0 μm, 2.85 × 10^8^ particles/m^3^ for particles 1.0–2.0 μm, 5.98 × 10^8^ particles/m^3^ for particles 2.0–5.0 μm, and 1.40 × 10^8^ particles/m^3^ for particles 5.0–10.0 μm. Thus, the increased particle size led to an 83.7% significant increase, 511.8% significant increase, 3044.7% significant increase, 5664.5% significant increase, and 1642.2% significant increase in the test area compared to the control area for each particle range, respectively (Figure 14).

On day 17, 6 June (65 days old), the aerosol particle concentrations were 2.71 × 10^8^ particles/m^3^ for particles 0.3–0.5 μm in diameter, 1.42 × 10^7^ particles/m^3^ for particles 0.5–1.0 μm, 7.50 × 10^6^ particles/m^3^ for particles 1.0–2.0, 1.11 × 10^7^ particles/m^3^ for particles 2.0–5.0 μm, and 7.76 × 10^7^ particles/m^3^ for particle 5.0–10.0 μm in the control area. On the other hand, in the test area, the concentration was 5.97 × 10^8^ particles/m^3^ for particles 0.3–0.5 μm in diameter, 3.48 × 10^7^ particles/m^3^ for particles 0.5–1.0 μm, 3.80 × 10^6^ particles/m^3^ for particles 1.0–2.0 μm, 4.96 × 10^6^ particles/m^3^ for particles 2.0–5.0 μm, and 3.45 × 10^6^ particles/m^3^ for particles 5.0–10.0 μm. There was a 120.4% significant increase, 144.5% significant increase, 49.3% significant decrease, 55.2% significant decrease, and 55.6% decrease in the test area compared to the control area for each particle ranges, respectively (Figure 15).

The observed concentration of aerosol particles demonstrated a more significant reduction in larger particles than smaller ones. This trend can be attributed to the larger diameter of these particles, which enhances their propensity for adhesion to the solution. The electrostatic adhesion force, known to be proportional to the diameter of the particle, further facilitates this adherence, particularly in the presence of ozone water [3]. Moreover, the sedimentation rate of particles, as delineated in Equation (2), suggests that larger aerosol particles are likely to settle more swiftly [3]. Additionally, a hypothesis is that the Optical Particle Sizer may have misidentified solution particles as aerosol particles due to their minute diameter.
(2)$$\begin{array}{c} V_{TS}=\frac{\rho_{p}d^{2}g}{18\eta } \end{array}$$
where $V_{TS}$ is the terminal settling velocity; $\rho_{p}$ is the density of particles; d is the diameter; g is the acceleration of gravity; η is the viscosity coefficient.

### 3.4. Aerosol MMass Concentration

The aerosol mass concentration was reduced for the test area compared to the control area on every experiment day. The average aerosol mass concentration was reduced by more than 50% in the test area compared to the control area, indicating that spraying the ozone water effectively reduced the aerosol mass concentration. Figure 16 shows the reduction rate of aerosol mass concentration in the test and control areas on days 4, 11, and 17.

On day 4, 22 May (52 days old), the average concentration was 4.6 mg/m^3^ in the control area and 2.2 mg/m^3^ in the test area; therefore, the average aerosol mass concentration reduced more than 56.0% in the test area compared to the control area. The average concentration on day 11, 29 May (59 days old) was 2.1 mg/m^3^ in the control area and 0.4 mg/m^3^ in the test area, which showed that the average aerosol mass concentration reduced more than 81.0% in the test area compared to the control area. For the reduction rate of aerosol mass concentration in the test and control area on day 17, 6 June (59 days old), the average concentration was 3.5 mg/m^3^ in the control area and 1.7 mg/m^3^ in the test area; thus, the average aerosol mass concentration reduced significantly by more than 51.0% in the test area. 

The aerosol mass concentration was reduced more than 50% in the test area compared to the control area. Since the filter used to make the measurements is completely dry of ozone water, the aerosol itself was reduced. This is because the solution attaches to the aerosol and falls to the floor; hence, the aerosol in the air was decreased in the test area compared to the control area. These measurements indicate that spraying the ozone water effectively reduced aerosol mass concentration.

## 4. Conclusions

This investigation aimed to assess the mitigation of aerosol particle concentrations, airborne microbial concentrations, and airborne bacteria through spraying an ozone solution. The experiments confirmed that spraying an ozone solution using an ultrasonic sprayer can mitigate the indexed factors and help decrease the risk of airborne infection for weanling pigs. These results underscore the effectiveness of ozone solution spraying to mitigate the spread of pathogens via aerosols and suggest that it is a promising strategy to improve sanitation in a livestock barn, rather than the traditional “all in/all out” sanitation. In prior research, the use of oil-based spraying methods was found to reduce airborne aerosols, yet achieving a consistent reduction required the use of a greater amount of oil [32]. However, the system introduced in this study could directly generate ozonated water from tap water and successfully reduce aerosols, which means this method also has economic advantages.

This alternative strategy holds the potential to increase both livestock biosecurity and productivity. In conclusion, the findings of this study could be pivotal for enhancing public health within piggery barns and contribute significantly to the exploration of effective methods for controlling aerosol particles and various airborne microbes in these environments. Ultimately, the primary theoretical implications of this research extend to the reduction in environmental pollution and the transmission of infectious diseases emanating from piggery barns, offering substantial benefits to the environment and public health.

## Figures and Tables

**Figure 1 animals-14-00657-f001:**
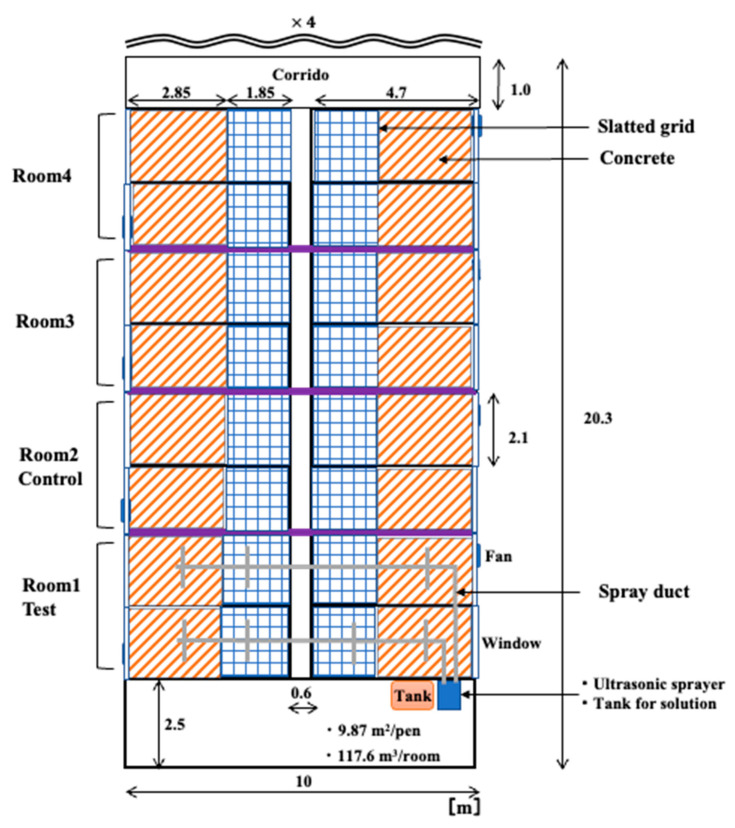
The plan of inside the experimental farm.

**Figure 2 animals-14-00657-f002:**
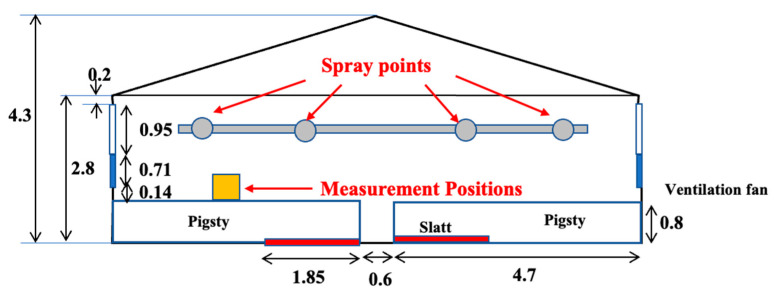
The measurement and spray points.

**Figure 3 animals-14-00657-f003:**
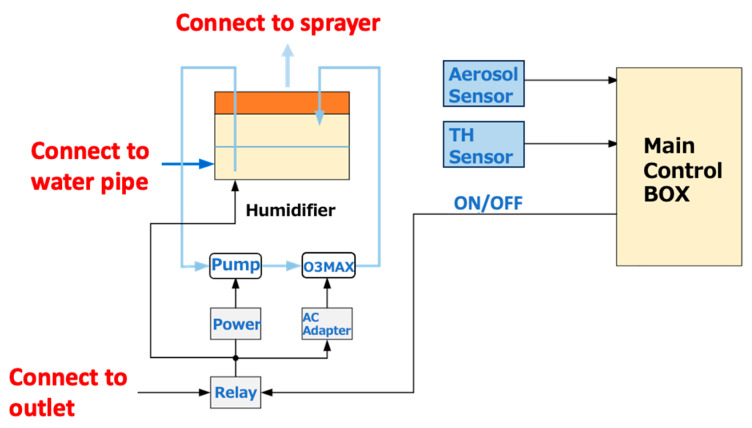
The spraying system employed in this experiment.

**Figure 4 animals-14-00657-f004:**
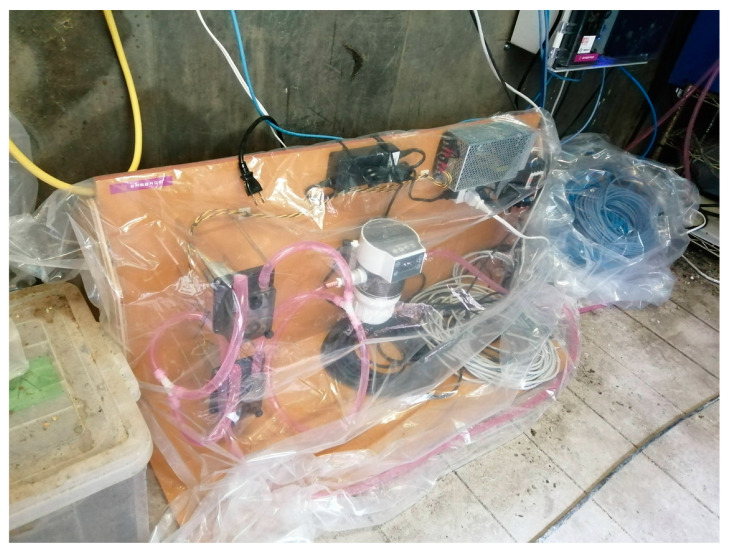
The photograph of the system prior to installation.

**Figure 5 animals-14-00657-f005:**
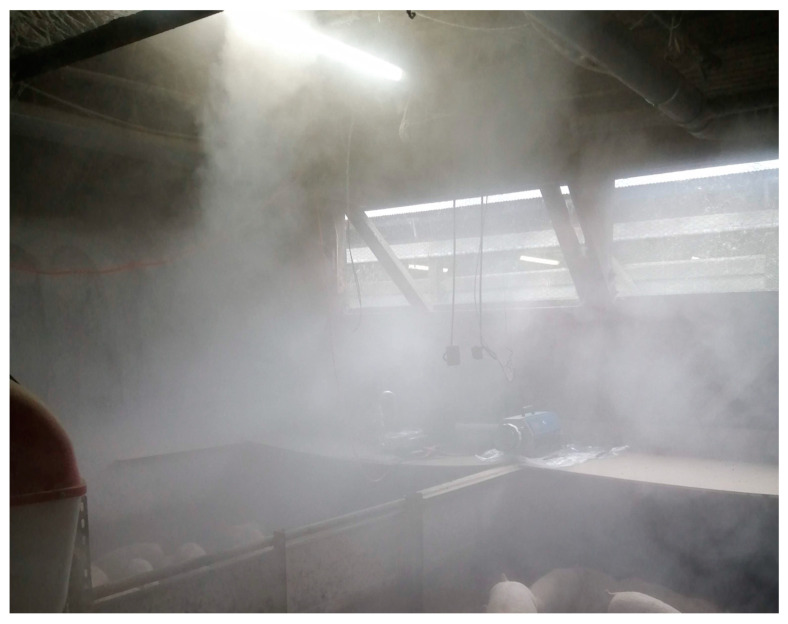
The image during spraying.

**Figure 6 animals-14-00657-f006:**
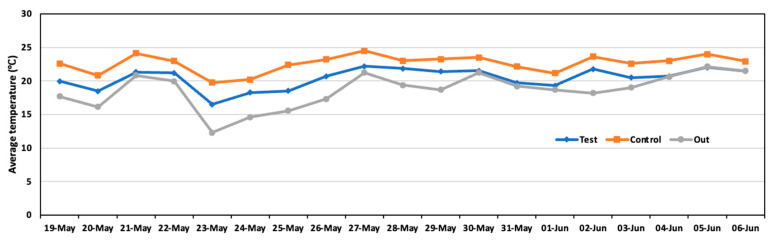
Daily average temperature in the test area, control area, and outside during the experiment.

**Figure 7 animals-14-00657-f007:**
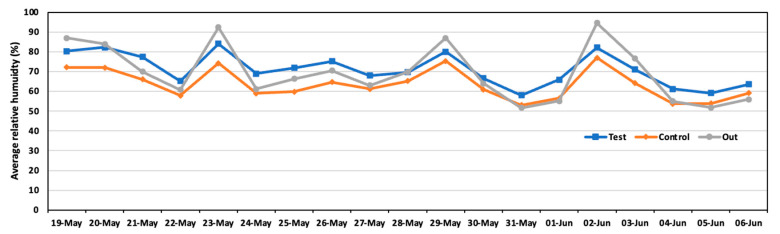
Daily average relative humidity in the test area, control area, and outside during the experiment.

**Figure 8 animals-14-00657-f008:**
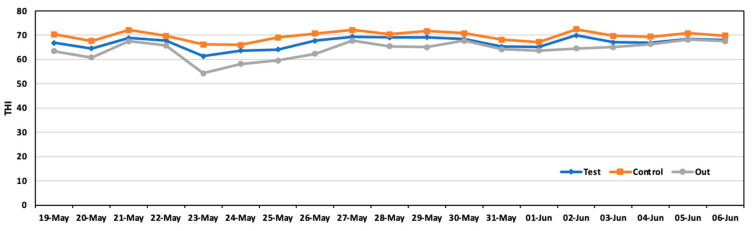
Daily average THI in the test area, control area, and outside during the experiment.

**Figure 9 animals-14-00657-f009:**
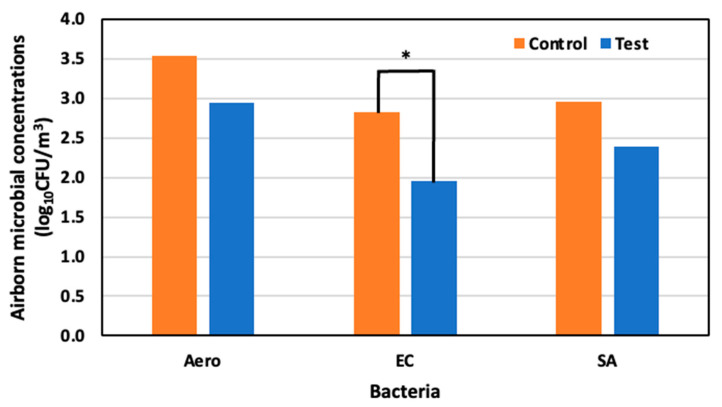
Airborne microbial concentration in the control and test area on day 4 at 52 days old; *: *p* < 0.05.

**Figure 10 animals-14-00657-f010:**
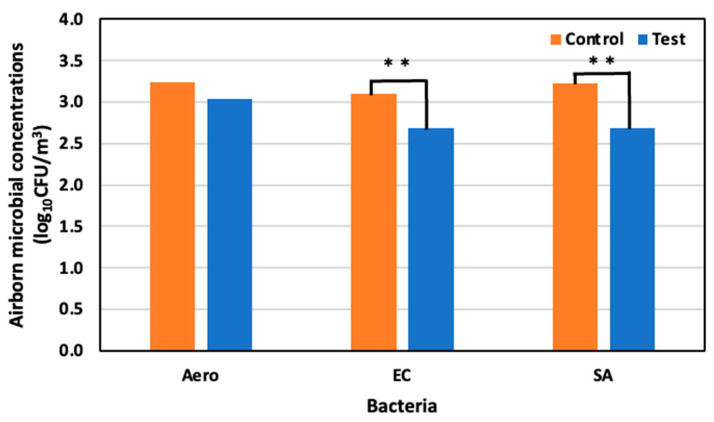
Airborne microbial concentration in the control and test areas on day 11 at 59 days old; **: *p* < 0.01.

**Figure 11 animals-14-00657-f011:**
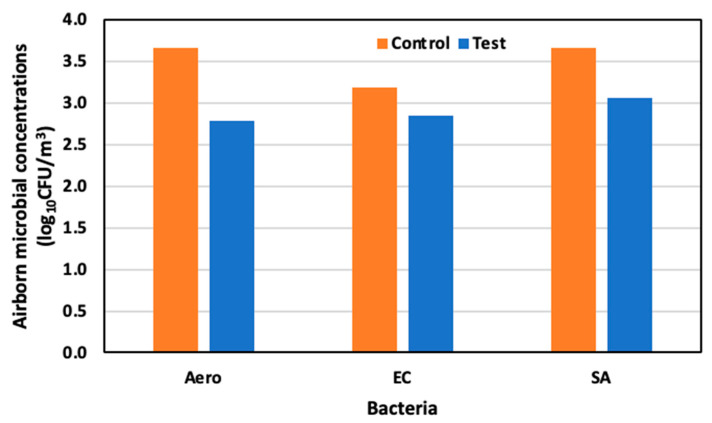
Airborne microbial concentration in the control and test area on day 17 at 65 days old.

**Figure 12 animals-14-00657-f012:**
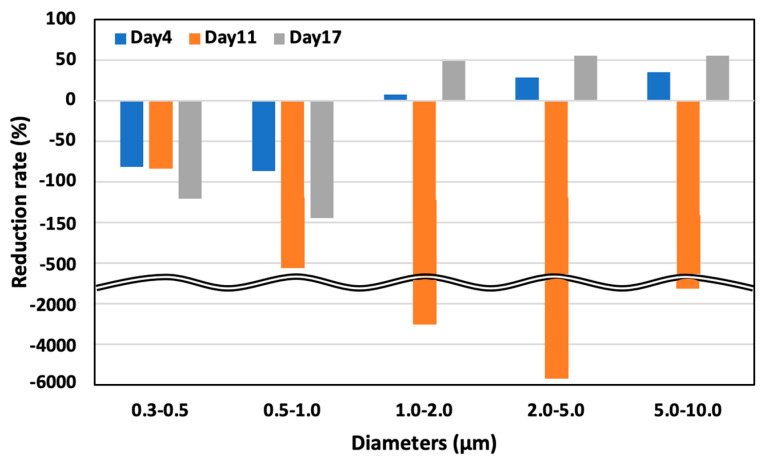
Aerosol particle concentration in the test area on days 4, 11, and 17.

**Figure 13 animals-14-00657-f013:**
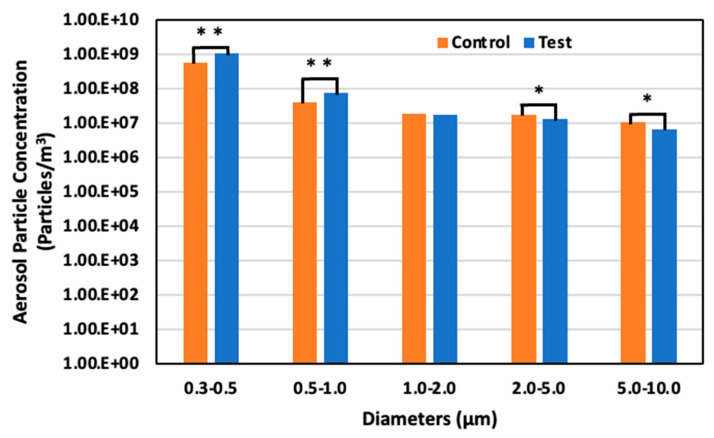
Aerosol particle concentration in the control and test areas on day 4 at 52 days old; *: *p* < 0.05; **: *p* < 0.01.

**Figure 14 animals-14-00657-f014:**
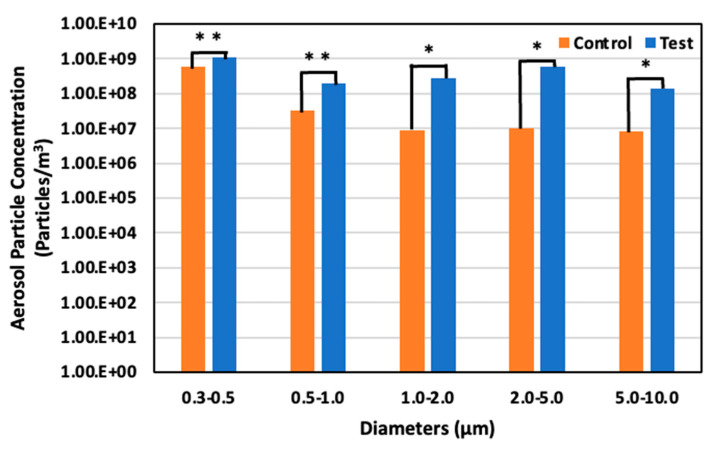
Aerosol particle concentration in the control and test areas on day 11 at 59 days old; *: *p* < 0.05; **: *p* < 0.01.

**Figure 15 animals-14-00657-f015:**
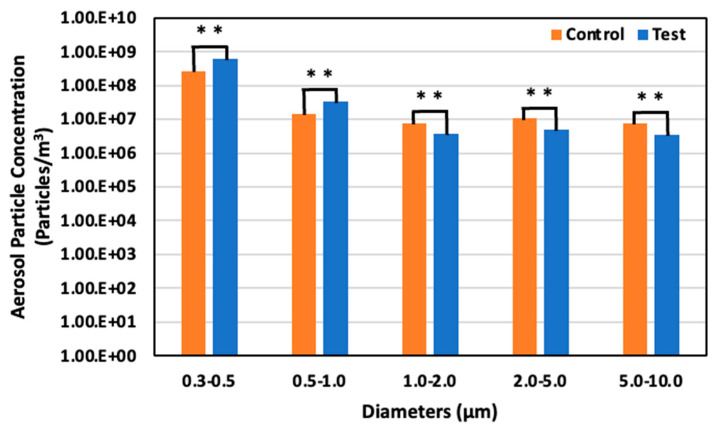
Aerosol particle concentration in the control and test areas on day 17 at 65 days old; **: *p* < 0.01.

**Figure 16 animals-14-00657-f016:**
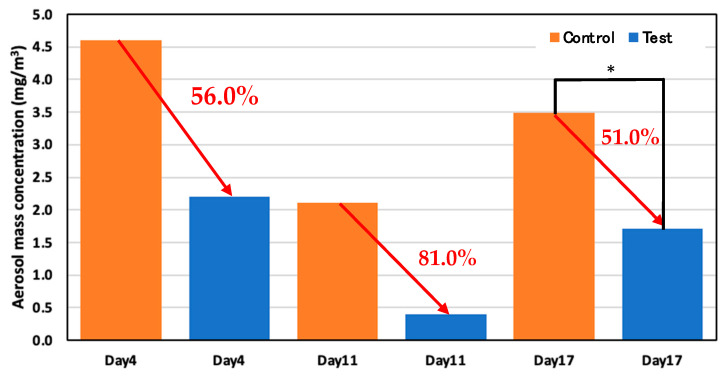
Aerosol mass concentration in the control and test areas on days 4, 11, and 17; *: *p* < 0.05.

## Data Availability

The data are contained within the article.

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
