# Peer review of "Mitigation of Aerosol and Microbial Concentration in a Weaning Piggery by Spraying Nanobubble Ozone Water with an Ultrasonic Sprayer"

_animals, 2024, doi:10.3390/ani14050657_

Round 1
Reviewer 1 Report
Comments and Suggestions for Authors
The authors studied mitigation of aerosol particles and airborne microbial concentrations using an ultrasonic sprayer to spray ozone solutions. The manuscript needs some extensive revisions before being accepted for publication. Here are some suggestions for the authors to consider. Thank you.
Clearly described objective(s) should be mentioned near the end of the Introduction.
Although ozone can be germicidal, it is also harmful to animals and workers. The author should describe whether the ozone concentration was within the safety and health threshold. If the ozone level is excessive, this application may face obstacles in implementation in farms.
The residence time of the sprayed aerosol suspending in the air is crucial to the bacteria mitigation objective. The authors should provide information related to that. Also, evaporation of aerosol is also very common, especially in summer times, which also plays a role in the effect of the application.
Line 149, how many experimental trials were conducted and how many replications?
Figure 4 and Figure 5, x-axis can be simplified by removing 2023 because they are all in the same year.
Figure 13 seems unnecessary as the information it presents can simply be described in a sentence.
Figure 14 – 16 should be combined into one figure to make the layout more organized to the reader.
Some on-site photos can be added to showcase the sprayer application process and make the manuscript more informative to the readers.
Comments on the Quality of English LanguageProofreading is needed to correct grammatical errors. For example, line 92 has errors. Line 149 has errors, “there were conducted...” is incorrect. There are more and the authors should fix them.
Reviewer 2 Report
Comments and Suggestions for Authors
This is an article with a topic that fits the scope of Animals. The study investigated an ultrasonic spraying system by evaluating the aerosol and microbial concentrations in the weaning piggery. Generally, the experiment is well designed and the approach is appropriate. However, by reading the full manuscript, I found many basic issues regarding the writing, data processing and data presentation. The writing definitely needs improvement. Some parts were written carelessly as there were errors/mistakes unacceptable for a peer-reviewed publication. Therefore, I would recommend a major revision at this point, and I would like the authors to seriously revise the context and address the questions, according to the comments below.

Comments on the Quality of English LanguagePlease double check the context's grammar carefully. There are some sentences need revising as pointed out in the comments. Also, I recommend the author not use some words that rarely used in scientific papers. Maybe it is a good idea to ask for some English writing assistance to help improve the manuscript.
Reviewer 3 Report
Comments and Suggestions for Authors
The manuscript “Mitigation of Aerosol and Microbial Concentration in the Weaning Piggery by Spraying the Nano Bubble Ozone Water with Ultrasonic Sprayer” is interesting.
Maintaining a hygiene environment in the livestock housing is very important for productivity improvement of livestock. Bioaerosols act as vectors for airborne pathogens and there is a strong correlation between microbe and aerosol particles. Therefore, the effective mitigation of aerosol particles is required for reduction in microbial populations. Some technologies like electrostatic precipitator, spraying weakly acidic water, photocatalytic ventilation system are available to decline the aerosol and microbe inside of livestock housing. The current manuscript focused on reducing concentrations of aerosol particles, airborne microbial concentrations, and airborne mass concentrations by spraying ozone solution with ultrasonic sprayer. Overall the work is important and relevant. Below are some comments in the spirit of helping the authors improve the manuscript.
1. The introduction section is well organized. However, gaps in knowledge statement is missing. Kindly mention gaps in knowledge. Moreover, mention the major objective of the study in introduction section.
2. ‘There were 80 pigs in the test area and 88 pigs in control area which was 49 years old when the research has started’. I didn’t get. Which was 49 years old? Kindly clarify.
3. How ‘Nano bubble ozone water’ was prepared and characterized? Kindly give details
4. E. coli, and Staphylococcus aureus etc. should be in italics
5. Statistical analysis is missing in M & M section. Kindly give details
6. In results, the authors mentioned temperature and relative humidity. Kindly calculate temperature-humidity index (THI) as it represents the combined effects of air temperature and humidity.
7. The authors are requested to strengthen the discussion with few more related literature
Comments on the Quality of English LanguageModerate editing is required.
Reviewer 4 Report
Comments and Suggestions for Authors
Firstly, I would like to congratulate the authors of the manuscript on their very interesting study on “Mitigation of Aerosol and Microbial Concentration in the Weaning Piggery by Spraying the Nano Bubble Ozone Water with Ultrasonic Sprayer”. The study aimed at highlighting the importance of future research evaluating, through decreasing aerosol and microbial concentrations, along with airborne viruses, by employing ozone solution spraying.
The manuscript reveals valuable information however, it lacks a major section of the manuscript that is statistical analyses. The results section that is not supported by the statistical analyses lacks its strength. The authors are requested to add a statistical analyses section, as well as define the statistical model properly with all the data presented as figures in the result section.
In my opinion the manuscript should be accepted for publication with observation that the authors should address before final submission.
1. L 27 Define S Farm.
2. Arrange keywords alphabetically.
3. L 106 format reference, Naide et.al (2018) [18]
4. L 111-112 format references, by Takehana et.al [17]
5. References need to be formatted as per journal guidelines throughout the paper please.
6. The experimental unit needs to be explained properly in the methodology and results
section.
7. The Statistical Analysis section is missing.
8. The results section is not supported by a strong statistical explanation.
9. Major academic editing needed before final submission.
10. Figure 1 There are four rooms whereas Room 1 and 2 are designated for test and control respectively however the Room 3 and 4 are not designated to test or control?
Best of Luck.
Comments on the Quality of English LanguageMajor academic editing needed before final submission.
Best of Luck.
Round 2
Reviewer 1 Report
Comments and Suggestions for Authors
Thank you for your effort in improving this manuscript. There are a number of grammatical errors throughout this manuscript that need to be fixed before publication.
Comments on the Quality of English LanguageLine 146: should be "equipment"
Line 155: should be "Figure 5 shows..."
And in other lines, there are too many to be listed here...
Author Response
I appreciate the detailed feedback provided, which has been instrumental in enhancing the clarity and coherence of the manuscript. I changed every revised and corrected grammar and word.Thank you.
Reviewer 2 Report
Comments and Suggestions for Authors
The manuscript has been significantly improved since the first round of review. I appreciate the effort authors have made to correct all the mistakes and address the questions raised by the reviewers.
I have a minor comment/question for the authors: what are the double curve lines representing in Figure 12? I am a bit confused.
Comments on the Quality of English Language
After the revision, the English writing has been improved.
Author Response
I appreciate the detailed feedback. For answering the question, in this Figure 12, a portion of the vertical axis has been omitted, and this section is indicated with a wavy line.Thank you.
Reviewer 3 Report
Comments and Suggestions for Authors
The manuscript has been significantly improved and the authors have addressed all the points raised by me. The current form of the manuscript looks very good and I feel it should be accepted for publication.
Comments on the Quality of English LanguageMinor editing is required.
Author Response
I appreciate your feedback. I updated my sentence again. Thank you.
Reviewer 4 Report
Comments and Suggestions for Authors
The revised version is lot better than earlier however, there are still some minor English academic editing needed.
Line 27 Replace S farm with 5 farm.
Line 146 Replace equioment with equipment.
Similarly, the authors needs to check punctuation marks throughout manuscript.
Comments on the Quality of English LanguageThe revised version is lot better than earlier however, there are still some minor English academic editing needed.
Line 27 Replace S farm with 5 farm.
Line 146 Replace equioment with equipment.
Similarly, the authors needs to check punctuation marks throughout manuscript.
Author Response
I appreciate the detailed feedback provided, which has been instrumental in enhancing the clarity and coherence of the manuscript. Each suggestion was carefully considered and incorporated to improve the overall quality of the work. Thank you.